# Imaging the Brainstem Raphe in Medication-Overuse Headache: Pathophysiological Insights and Implications for Personalized Care

**DOI:** 10.3390/biomedicines13010131

**Published:** 2025-01-08

**Authors:** Annika Mall, Christine Klötzer, Luise Bartsch, Johanna Ruhnau, Sebastian Strauß, Robert Fleischmann

**Affiliations:** Department of Neurology, University Medicine Greifswald, 17489 Greifswald, Germanyjohanna.ruhnau@uni-greifswald.de (J.R.);

**Keywords:** migraine, medication-overuse headache, chronic migraine, addiction, brainstem raphe, transcranial ultrasound, serotonergic system, biomarker, precision medicine

## Abstract

**Background/Objectives**: Medication-overuse headache (MOH) is a disabling condition affecting patients with chronic migraine resulting from excessive use of acute headache medication. It is characterized by both pain modulation and addiction-like mechanisms involving the brainstem raphe, a region critical to serotonergic signaling. This study investigates whether alterations in the brainstem raphe, assessed via transcranial sonography (TCS), are associated with MOH and independent of depressive symptoms, aiming to explore their utility as a biomarker. **Methods**: This prospective case-control study included 60 migraine patients (15 with MOH) and 7 healthy controls. Comprehensive clinical and psychometric assessments were performed to evaluate headache burden, medication use, and depressive symptoms. TCS was used to assess brainstem raphe echogenicity, with findings analyzed using generalized linear models adjusted for depression. **Results**: Non-visibility of the brainstem raphe was significantly associated with MOH, with an unadjusted odds ratio (OR) of 6.88 (95% CI: 1.32–36.01, *p* = 0.02). After adjusting for depressive symptoms, this association remained significant, with an adjusted OR of 1.85 (95% CI: 1.02–3.34, *p* = 0.041). TCS demonstrated good intraclass correlation, highlighting its reproducibility and ability to detect changes relevant to MOH pathophysiology. **Conclusions**: Brainstem raphe alterations are associated with MOH and may serve as a potential biomarker for its diagnosis and management. TCS offers a non-invasive, cost-effective tool for identifying MOH-specific mechanisms, which could improve clinical decision-making and support personalized care in chronic headache disorders. Further studies are needed to validate these findings and refine the clinical applications of brainstem-focused diagnostics.

## 1. Introduction

Medication-overuse headache (MOH) represents a significant public health concern, burdening patients with profound challenges and exacerbating the overall societal impact of chronic pain disorders [1]. MOH arises from the frequent use of symptomatic headache medications, paradoxically resulting in the progression and chronification of primary headache disorders such as migraine [2]. The diagnosis of MOH, as per the International Classification of Headache Disorders (ICHD-3), requires headaches on ≥15 days per month due to the overuse of acute medications for ≥3 months [3]. Affecting an estimated 0.5% to 2% of the global population, MOH not only deteriorates the quality of life but also complicates treatment pathways for those affected [4].

Patients with MOH frequently exhibit refractoriness to standard headache management protocols, necessitating innovative therapeutic strategies. Current approaches often rely on general preventive migraine medications, sometimes coupled with detoxification protocols aimed at medication withdrawal [4]. While these strategies can yield clinical improvement, they lack specificity to the underlying pathophysiology of MOH. Consequently, MOH relapses are common, and their sustained resolution remains challenging [5]. Addressing this gap necessitates a deeper understanding of the central mechanisms contributing to MOH pathogenesis.

Emerging evidence suggests that alterations in central reward and pain modulation systems may underlie both MOH and behaviors associated with substance dependency [4]. The brainstem raphe, a midline structure rich in serotonergic neurons, is integral to these functions, playing a dual role in modulating nociception and influencing reward pathways [6]. In the context of pain modulation, serotonergic pathways from the raphe nuclei regulate ascending and descending nociceptive signals, directly affecting pain-sensitive neurons in the thalamus and spinal dorsal horn [7]. Concurrently, addiction-like behaviors are characterized by disruptions in the same serotonergic signaling, which can lead to maladaptive neuroplastic changes [8]. These interlinked mechanisms involving both pain modulation and addiction-like processes form the foundation of the hypothesis that investigating brainstem raphe alterations is crucial to understanding the pathophysiology of MOH [7,9].

Neuroplastic alterations in the brainstem raphe are thus hypothesized to contribute to the persistence and progression of MOH [10]. Altered echogenicity of the brainstem raphe, detectable via transcranial sonography (TCS), has been observed in conditions such as depression, Parkinson’s disease, and migraine raphe ultrasound. Investigating the hypothesized interaction between brainstem raphe alterations and MOH is crucial to substantiate their pathophysiological relationship. TCS offers a non-invasive and widely available method to explore these changes, shedding light on MOH pathophysiology and potentially serving as a biomarker for the disorder.

Given the high prevalence of depressive symptoms in MOH and their potential to confound TCS findings, it is hypothesized that changes in the brainstem raphe are independent of depressive symptoms, which are also more frequently observed in patients with MOH compared to controls [11]. Depressive symptoms need to be controlled to test this hypothesis, especially since patients suffering from migraine frequently experience depressive symptoms.

This study tests the primary hypothesis that brainstem raphe alterations, as assessed through TCS, are detectable in MOH. Confirmation of this hypothesis would substantiate the relevance of associated pathways for the pathophysiology of MOH in the context of migraine. Finally, TCS might be a suitable diagnostic biomarker to inform decision-making in patients with chronic headaches by assessing the contribution of MOH-associated mechanisms.

## 2. Materials and Methods

### 2.1. Ethical Considerations

The study adhered to the ethical principles of the Declaration of Helsinki and received approval from the Institutional Review Board of University Medicine Greifswald, Germany (Approval Code: BB 161/18). All participants provided informed consent prior to participation, and no surplus material was utilized. Data collection and analysis were conducted using pseudonymized datasets to ensure confidentiality.

### 2.2. Study Design and Population

This prospective, exploratory case-control study was conducted at the outpatient headache clinic of University Medicine Greifswald. The study included sixty patients with a confirmed diagnosis of migraine according to the International Classification of Headache Disorders, 3rd edition (ICHD-3). Of these, fifteen patients were additionally diagnosed with medication-overuse headache (MOH) within the past five months. Seven healthy controls, matched for age and sex, were recruited to provide baseline reference data. Participants with other primary headache disorders, neurological diseases unrelated to migraine, or insufficient acoustic windows for transcranial sonography (TCS) were excluded.

### 2.3. Clinical and Psychometric Assessments

Comprehensive clinical and psychometric assessments were conducted to gather detailed information on headache history, medication use, and mental health status. The Migraine Disability Assessment (MIDAS) was used to evaluate the degree of disability caused by migraine over the previous three months, including its impact on work, household responsibilities, and social activities. The Headache Impact Test-6 (HIT-6) quantified the overall burden of headaches on daily life and quality of life, producing a score indicative of headache severity. The Depression, Anxiety, and Stress Scale (DASS) provided a validated assessment of psychological comorbidities, offering separate evaluations of depression, anxiety, and stress levels. The Veterans RAND 12-Item Health Survey (VR-12), adapted from the SF-36 health survey, was employed to measure health-related quality of life in both the physical and mental domains.

These tools enabled the collection of standardized data to control for potential confounders such as depression and anxiety, which are frequently comorbid in patients with migraine and MOH. The data provided a basis for understanding the broader impact of these conditions and supported efforts to isolate brainstem-specific changes in MOH.

### 2.4. Transcranial Sonography (TCS) Protocol

Transcranial sonography was used to evaluate the echogenicity of the brainstem raphe. Imaging was performed using a phased-array sector transducer with a frequency of 2.0–2.5 MHz. Parameters included an image depth of 13–16 cm and a dynamic range of 50–65 dB. Participants were positioned supine with their heads rotated 20–40 degrees to optimize the acoustic window. The temporal bone window was selected as the primary imaging approach. The accessibility of the temporal bone window, critical for effective TCS imaging, was considered during participant selection. Temporal bone acoustic windows are less accessible in older populations, particularly those over the age of sixty, due to increased bone density and calcification. This limitation was accounted for during the inclusion process to minimize potential biases related to imaging constraints.

The echogenicity of the brainstem raphe was assessed and compared to the red nucleus, which served as a reference structure. Echogenicity was scored on a three-point scale: 0 for no visibility of the raphe, 1 for reduced or discontinuous echogenicity, and 2 for continuous and well-defined echogenicity. Measurements of the raphe’s width and length were attempted at the widest points but were not included in the final analysis due to variability introduced by probe angle differences. Participants whose temporal windows did not allow visualization of either the red nucleus or the brainstem raphe were excluded.

### 2.5. Statistical Analysis

Statistical Package for the Social Sciences software (v29.0, IBM, Armonk, NY, USA) was used for all statistical evaluations. Descriptive data were presented, based on the distribution of the data, as frequencies, means with standard deviations, or medians with interquartile ranges. The primary endpoint, the difference in brainstem raphe echogenicity between patients with MOH and controls, was evaluated using a mixed model generalized linear model (GLM). Akaike Information Criterion (AIC) is a statistical measure used to compare and select models by evaluating how well they fit the data while penalizing for model complexity, thereby minimizing the risk of overfitting; in this study, AIC was applied to identify the GLM that provided the best balance between explanatory power and simplicity (with lower values indicating a better model). The rationale for the sample size calculation was based on the following parameters, using Cohen’s f as the effect size measure, which is appropriate for GLMs [12]. A medium effect size of Cohen’s f = 0.25 was selected. Unlike Cohen’s d, where d = 0.25 is considered a small effect size, f = 0.25 represents a medium effect. The calculation further assumed an alpha error probability of 0.05, a power of 0.80, three groups, and two measurements. These parameters yielded a required total sample size of 60, with an actual power of 0.8041753 (calculated using G*Power 3.19.2). This approach accounted for both longitudinal and cross-sectional data, utilizing an ordinal linking function appropriate for the scoring of echogenicity. The basic model was further expanded to evaluate secondary hypotheses, such as controlling for depression as a covariate, to explore whether the observed differences in echogenicity were independent of depressive symptoms.

Additionally, a subset of the longitudinal data was analyzed to investigate whether changes in TCS findings were associated with changes in the clinical phenotype, such as headache frequency or severity. This exploratory analysis aimed to assess dynamic relationships between brainstem raphe alterations and the progression or remission of MOH-related characteristics. A significance threshold of *p* < 0.05 was applied for all analyses, with data distribution assessed using Kolmogorov-Smirnov and Shapiro-Wilk tests.

## 3. Results

### 3.1. Patient Population and Headache Characteristics

A total of 67 participants were included in the study, comprising 60 migraine patients and 7 healthy controls. Among the migraine patients, 15 had a co-diagnosis of medication-overuse headache (MOH) within the last five months. The relapse rate among patients with MOH was 67% (*n* = 10). The mean age of the population was 41.5 ± 14.2 years in migraine patients without MOH (*n* = 45, 3 female), 44.7 ± 12.1 years in MOH patients (*n* = 15, 13 female), and 29 ± 1.5 years in healthy controls (*n* = 3 female). There was a numerical, yet statistically non-significant, larger proportion of patients with MOH receiving preventive medication (*p* = 0.14) and serotonin reuptake inhibitors (*p* = 0.08). Detailed patient characteristics are summarized in Table 1.

### 3.2. Association Between MOH and Brainstem Raphe Alterations

The generalized linear model for the association of MOH status and TCS findings was significant (F(2,71) = 3.33, *p* = 0.041). The model accuracy was 74% for the identification of MOH. The Akaike Information Criterion (AIC) of the model was 353. There were significant coefficients in the multinomial approximation of the ordinal distribution. A completely visible brainstem raphe was considered a normal finding, and a discontinuous and non-visible raphe was compared against it. A discontinuous raphe was not significantly increased with an increased probability of suffering MOH (beta = 0.51, *p* = 0.51, OR = 1.66 [95% CI: 0.36–7.64]). Invisibility of the brainstem raphe was significantly associated with the MOH status (beta = 1.93, *p* = 0.02, OR = 6.88 [95% CI: 1.32–36.01]). Representative images of TCS findings and their relationship with the model-based probability of MOH are shown in Figure 1. We explored whether reduced brainstem raphe continuity was potentially due to the larger proportion of chronic (CM) than episodic migraine (EM) in the MOH group given their higher headache frequency. Invisibility of the brainstem raphe was, however, not associated with CM across patients, irrespective of MOH (beta = 0.64, *p* = 0.29, OR = 1.89 [95% CI: 0.57–6.29]).

We then tested whether visibility of the brainstem raphe was still predictive of MOH when adjusted for depressivity. Since only the invisibility of the brainstem raphe was associated with the presence of MOH, we here treated the TCS findings as a binary predictor, i.e., with the brainstem raphe being either visible or not. Through this approach, binary logistic regression could be used to adjust TCS findings for DASS-D scores. The model was recalculated with adjusted TCS findings. The resulting GLM was more informative, indicated by an improved AIC of 313, and remained significant at *p* = 0.041. The brainstem raphe visibility adjusted for depressivity was associated with an OR_adj_ = 1.85 [95% CI: 1.02–3.34], i.e., non-visibility of the brainstem raphe was associated with about a two-fold likelihood of MOH. Patients with non-visibility of the brainstem raphe significantly more often received serotonin (norepinephrine) (SSRI/SSNRI) reuptake inhibitors (χ^2^(3) = 6.2, *p* = 0.045; OR = 5.7 [95% CI: 1.3–25.0]). The rate of preexisting major depression among MOH patients receiving SSRI/SNRI was not statistically significantly different (χ^2^(2) = 2.6, *p* = 0.11).

None of the control subjects had a non- or only partially visible brainstem raphe, which supports the methodological robustness and specificity of identified findings.

### 3.3. Longitudinal Changes in Brainstem Raphe and Clinical Phenotype

There were 8 patients with MOH (*n* = 2) and without MOH (*n* = 6), and repeated TCS measures. The intraclass correlation of brain stem raphe rating for the same clinical phenotype was 0.778, which was significant *p* = 0.03, i.e., findings were significantly correlated. There was one patient who presented with MOH at the time of measurement and had no visible brainstem raphe at that time. On follow-up three months later and following detoxification, brainstem raphe was continuously visible. These findings were not further statistically analyzed given the small sample size.

## 4. Discussion

This study provides evidence supporting the hypothesis that brainstem raphe alterations are associated with MOH in the context of migraine. Using transcranial sonography (TCS), we identified significant differences in the echogenicity of the brainstem raphe between patients with MOH, migraine patients without MOH, and healthy controls. These findings remained significant even after controlling for depressive symptoms, underscoring the independence of MOH-associated changes from commonly comorbid psychiatric conditions.

Our results align with the proposed role of the brainstem raphe as a critical modulator in nociceptive and reward pathways, both of which are implicated in MOH pathophysiology [13]. In addition, the dynamic nature of brainstem raphe alterations in a subset of patients with longitudinal data suggests a potential link between these changes and the clinical phenotype of MOH.

### 4.1. Pathophysiological Implications

The findings contribute to the understanding of MOH pathophysiology, supporting the involvement of central mechanisms in chronic headache disorders. The brainstem raphe, a structure densely populated with serotonergic neurons, plays a pivotal role in regulating both ascending and descending nociceptive signals. Serotonergic dysfunction has been implicated in MOH pathogenesis, and this study extends the evidence to MOH by demonstrating structural and functional brainstem abnormalities [14].

The alterations in raphe echogenicity observed in MOH patients may reflect neuroplastic changes induced by chronic pain and medication overuse [15]. Similar neuroplasticity is evident in other chronic pain conditions and substance dependency, where maladaptive changes in reward and pain modulation circuits perpetuate pathological behaviors [16]. These findings are consistent with previous imaging studies reporting volumetric brainstem changes in MOH and the overlapping pathophysiological mechanisms between MOH and addiction-like behaviors [17].

Furthermore, the presence of significant brainstem raphe alterations independent of depressive symptoms strengthens the specificity of these changes to MOH-related mechanisms. While serotonergic dysfunction is central to depression, the unique pattern of raphe changes in MOH underscores its distinct pathophysiological profile, emphasizing the need for targeted therapeutic approaches.

### 4.2. Clinical Implications

The clinical consequences of these findings are substantial. Current management of MOH typically relies on general migraine preventive therapies which may be combined with detoxification protocols, which have variable success rates and high relapse rates [18]. The identification of brainstem raphe alterations as a pathophysiological marker of MOH could inform more targeted interventions. In the future, behavioral treatment approaches useful in treating substance addiction might be suitable based on the current findings [16].

Furthermore, in cases with insufficient response to migraine-specific treatment, it may be challenging to decide whether the preventive treatment of migraine requires a switch or, alternatively, whether MOH is present and might contribute to the headaches. Thus, integrating TCS into routine clinical practice offers a non-invasive and cost-effective tool for identifying patients at risk of MOH and informing clinical decision-making. The potentially dynamic nature of raphe alterations further suggests its potential utility in assessing treatment efficacy over time. There is clear evidence that, in depression, brainstem raphe alterations may be suitable to monitor disease activity [19]. Our study provides limited evidence in that direction, paving the way for personalized medicine in chronic headache management, which requires further validation in prospective studies.

Interestingly, brainstem raphe invisibility was associated with the presence of MOH but not CM. Our results align with previous studies showing that approximately 20% of patients with EM but not CM may suffer MOH [20]. These patients exhibited similar alterations in brainstem raphe morphology on TCS in our study, suggesting a distinct finding linked specifically to MOH rather than migraine type.

### 4.3. Utility of TCS Findings as Biomarkers

The findings from this study highlight the potential of TCS-derived brainstem raphe alterations as a biomarker for MOH. Echogenicity changes detected via TCS provide a direct, real-time measure of structural abnormalities in the brainstem, which are closely tied to MOH pathophysiology. Unlike clinical criteria that rely on subjective reporting, TCS offers objective data, which could enhance diagnostic accuracy and treatment stratification [21]. Exploratory analyses suggest that brainstem raphe echogenicity may reflect the efficacy of SSRI/SNRI treatment in patients with MOH. Patients with non-visible brainstem raphe were more likely to be prescribed serotonergic medications, indicating that TCS could help identify individuals who respond to treatments targeting serotonergic pathways, as seen previously in major depression [19]. While no significant differences in MOH prevalence were observed between groups receiving serotonergic medications, patients with MOH and reduced brainstem raphe visibility appeared to benefit from continued SSRI/SNRI treatment beyond what could be explained by preexisting depression. This supports the potential role of TCS in guiding treatment decisions for headache patients. Similarly, another study found that the Intensity-Dependence of Auditory-evoked Potentials (IDAP), a marker of central serotonergic tone, was steeper in chronic migraine patients, many of whom had MOH, reflecting lower serotonergic activity [22]. Treatment with greater occipital nerve blocks normalized IDAP signaling, suggesting that serotonergic pathways can improve with targeted interventions and TCS of the brainstem raphe may likewise aid in tailoring treatment strategies. However, this treatment does not need to primarily directly target serotonin but rather CM and MOH, as suggested by current guidelines. Finally, the widespread availability of TCS and its non-invasive nature make it an attractive option for longitudinal studies and routine clinical applications. By incorporating TCS into the diagnostic workflow, clinicians could better differentiate MOH from other chronic headache disorders and assess the impact of therapeutic interventions on brainstem function.

### 4.4. Limitations

Despite the promising findings, this study has several limitations. The sample size, particularly for the longitudinal subsets, was relatively small, limiting the generalizability of the results. Future studies with larger cohorts are needed to validate these findings and establish robust normative data for brainstem raphe echogenicity.

Another limitation is the cross-sectional design for the majority of the study population, which precludes causal inferences about the relationship between brainstem raphe alterations and MOH. Nonetheless, adjusting the findings for psychiatric comorbidity changed the effect size but not the independent contribution of TCS findings to MOH.

Additionally, the variability in acoustic window quality, particularly in older participants, could introduce measurement bias [23]. While efforts were made to optimize imaging conditions, this remains a potential confounder. Lastly, although depressive symptoms were controlled for in the analysis, other psychiatric and behavioral factors may also influence brainstem raphe echogenicity and warrant further investigation.

## 5. Conclusions

This study establishes that brainstem raphe alterations, detected via TCS, are associated with MOH in migraine patients, independent of depressive symptoms. These findings highlight the role of serotonergic dysfunction in MOH pathophysiology and suggest a potential overlap with mechanisms underlying chronic pain and maladaptive behaviors.

TCS emerges as a promising biomarker, offering a non-invasive, cost-effective tool for diagnosing and monitoring MOH. While further research is needed to validate these findings and refine clinical applications, this study underscores the utility of brainstem-focused approaches in advancing MOH management and personalized headache care.

## Figures and Tables

**Figure 1 biomedicines-13-00131-f001:**
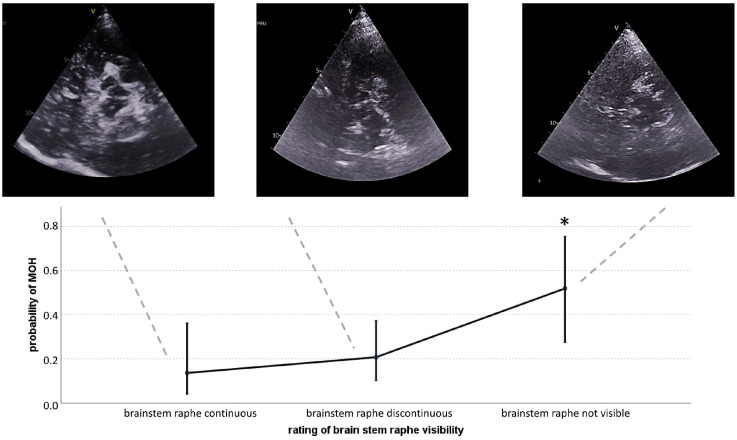
Illustration of TCS findings in comparison to MOH probability in the study sample. While any visibility of the brainstem raphe (i.e., continuous or discontinuous) was associated with rather low probabilities of MOH, an invisible brainstem raphe increased the odds of suffering MOH. Grey dashed lines indicate the visibility category to which the respective figure belongs. Solid lines represent the standard deviations of the intervals of the mean probability. * = statistically different from continuous raphe.

**Table 1 biomedicines-13-00131-t001:** Patient characteristics including headache features and patient-reported outcomes. None of the patient characteristics were statistically different between groups with and without MOH, except for headache frequency. DASS = Depression, Anxiety, and Stress Scale. SSRI/SSNRI = selective serotonin (norepinephrine) reuptake inhibitor. * = significantly different between groups.

	Migraine	
	With MOH (*n* = 15)	Without MOH (*n* = 45)
Headache frequency	18.4 ± 9.8 *	10.6 ± 8.0
Fulfilling criteria of chronic migraine (%)	79% *	44%
Analgesics per month	18.4 ± 9.3 *	5.3 ± 3.1
Current preventive medication (%)	84%	67%
Current SSRI or SSNRI (%)	31%	12%
DASS-D	13.3 ± 10.3	9.9 ± 9.5
DASS-A	13.2 ± 10.1	8.3 ± 8.4
DASS-S	17.3 ± 8.2	12.8 ± 9.3
VR-12 PCS	34.6 ± 5.5 *	40.1 ± 9.7
VR-12 MCS	37.2 ± 11.5	41.3 ± 12.7
MIDAS	92.9 ± 63.4 *	44.1 ± 47.7
HIT-6	64.3 ± 2.9 *	60.1 ± 7.9

## Data Availability

Data available on request due to restrictions regarding data protection regulations.

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
