# Peer review of "Imaging the Brainstem Raphe in Medication-Overuse Headache: Pathophysiological Insights and Implications for Personalized Care"

_biomedicines, 2025, doi:10.3390/biomedicines13010131_

Round 1
Reviewer 1 Report
Comments and Suggestions for Authors
The paper by Dr. Annika Mall et al deals with the investigation on the possible diagnostic role of Doppler ultrasound of the brainstem serotonergic structures in discriminating among patients with Medciation overuse Headache (MOH) vs. patients with episodic migraine (EM and healthy control (HC). The paper is very interesting and I think that it could be very promising in the field of migraine and MOH. The non-invasive investigation of parenchymal brain structures by duplex is providing interesting evidence in several fields of neurology.
The paper is overall well written and clear in any part of the manuscript.
I have some point I would like to ask authors for a clarification with the hope that these points can help to increase the robustness of the paper.
One first point that needs some clarification is the referring to Akaike Information Criterion (AIC) in the results without having explain what it is in the methods part. While some research has an idea of what AIC is, many could be confused by the abrupt dumping of this term, especially because it is very briefly explained. I think that a deeper explanation of what it is and in which analysis it is used can makes the text clearer by the general audience.
Another question is about recruitment, the patients included in the paper were naïve to treatment? Or were they patients with previous relapses in MOH? Were they devoid from preventive therapy? As well, were serotonergic medications used for inducing the remission from MOH to episodic form during the follow-up phase? Some preventive therapy could modify serotonergic transmission and possibly represents a bias.
Statistically, I would like to ask some questions. First, why authors used a GLM? As far as I understand, input variables are categorical as raphe presence categorized by 0 (absence), 1 (incomplete), 2 (complete) visualization of the raphe. I wonder whether a more direct chi-square or Fisher-Freeman-Halton’s test could serve better the analysis. Maybe some multivariate approach as discriminant function analysis could be useful but I guess that number of patients could easily leads to overfitting. Secondly, I think that defining 0.25 as a medium effect is a little misleading since 0.2 is generally considered a small effect (with therefore a low level of generalizability of the results). I suggest authors to discuss this point in the limits of the paper. As third point, I wonder why authors have used to normality test while just one is enough to determine the distribution of data.
Another point that should be addressed in my opinion is that authors divided by patients with MOH and therefore chronic migraine + MOH and patients with 10 days per month of overall headache (thus episodic migraine). I think that the more robust comparison would have been with MOH-CM vs. CM without MOH. As it is, the paper cannot exclude that the lack of visualization of raphe is indeed a sign of MOH or a sign of chronicity. In this sense the longitudinal part of the study is not helpful since remission from MOH also implies remission from CM to episodic migraine.
The discussion is well-written and interesting. I think that results presented are very promising and are in line with previous literature showing that chronic migraine patients (in large share MOH patients) have an increase of serotonin firing recorded by IDAP from brainstem to cortex when successfully treated with GON block from very disabling form to episodic form. And more interesting the size of the serotonin increase correlates directly with the reduction of headache days. In this sense, I would suggest authors to expand the discussion stressing that their results fit in a larger context.
Comments on the Quality of English LanguagePlease check some typos.
Reviewer 2 Report
Comments and Suggestions for Authors
Annika Mall and colleagues submitted a research manuscript titled, "Imaging the Brainstem Raphe in Medication-Overuse Headache: Pathophysiological Insights and Implications for Personalized Care" to Biomedicines. The authors conducted a prospective case-control study involving 60 migraine patients, including medication-overuse headache (MOH) patients (15) and healthy volunteers (07). The authors performed clinical and psychometric assessments, and transcranial sonography (TCS) was used to assess brainstem raphe echogenicity. Authors identified that TCS provided good intraclass correlation, reproducibility and detection of minor changes in MOH pathophysiology. Authors concluded that Brainstem raphe alterations parameters could be potential biomarkers for diagnosis and management of MOH.
This is a good concept-driven research work demonstrating the improvement of clinical decision-making and personalized care in chronic headache disorders.
The introduction part covered the fundamental information related to the topic.
The figure and table are presented clearly and the methodology covered the required details related to the experimentation.
The conclusion is well aligned with the objective of the undertaken study.
Authors also provided the limitations of the present study and the future direction of the proposed research study.
Following minor revision is needed to the manuscript:-
1. References must be updated according to the recently published literature, wherever applicable.
2. References need to be included in 3rd paragraph of Introduction.
3. Section 2.5: SPSS (accession date needs to be mentioned).
4. Section 3.2: Akaike information criterion (AIC), authors should explain this parameter, what is the significance of AIC?
Round 2
Reviewer 1 Report
Comments and Suggestions for Authors
Authors have fulfilled my comments and suggestions. I endorse the publication of the paper in the present form.